# Co-Pyrolysis of Unsaturated C4 and Saturated C6+ Hydrocarbons—An Experimental Study to Evaluate Steam-Cracking Performance

**DOI:** 10.3390/ma16041418

**Published:** 2023-02-08

**Authors:** Jiří Petrů, Tomáš Herink, Jan Patera, Petr Zámostný

**Affiliations:** 1Department of Organic Technology, Faculty of Chemical Technology, University of Chemical Technology Prague, Technická 5, Dejvice, 166 28 Prague, Czech Republic; 2Unipetrol RPA, Záluží 1, 436 70 Litvínov, Czech Republic; 3University Center Litvínov ICT-FS ČVUT-ORLEN Unipetrol, Záluží 1, 436 70 Litvinov-Záluží, Czech Republic

**Keywords:** pyrolysis, co-pyrolysis, C4 unsaturated hydrocarbons, butenes, steam cracking

## Abstract

Unsaturated C4 hydrocarbons are abundant in various petrochemical streams. They can be considered as a potential feedstock for the steam-cracking process, where they must be co-processed with C6 and higher (C6+) hydrocarbons of primary naphtha fractions. Co-pyrolysis experiments aiming at the comparison of different C4 hydrocarbon performances were carried out in a laboratory micro-pyrolysis reactor under standardized conditions: 820 °C, 400 kPa, and 0.2 s residence time in the reaction zone. C4 hydrocarbons were co-pyrolyzed with different co-pyrolysis partners containing longer hydrocarbon chain to study the influence of the co-pyrolysis partner structure on the behavior of C4 hydrocarbons. The yields of the pyrolysis products and the conversion of C4 hydrocarbons were used as the performance factors. A regression model was developed and used as a valuable tool for quantifying the inhibition or acceleration effect of co-pyrolysis on the conversion of co-pyrolyzed hydrocarbons. It was found that the performance of different C4 hydrocarbons in co-pyrolysis is substantially different from the separate pyrolysis of the individual components.

## 1. Introduction

Steam cracking is one of the most significant processes in the petrochemical industry. At present, mostly petroleum fractions and streams from various petrochemical and refinery processes are subjected to steam cracking, with the aim of obtaining very important intermediates for the petrochemical industry, particularly hydrogen, ethylene, propylene, butadiene, toluene, and xylenes, which are consequently used as monomers in the production of polymeric materials. Along with these hydrocarbons, C4 fraction represents another important product of the steam-cracking process [1]. Except for the steam-cracking process, the second source of C4 fraction is refinery fluidized catalytic-cracking process. Both the steam-cracking and fluid catalytic-cracking processes are standardly integrated in refinery–petrochemical complexes to maximize feedstock and the utilization of products. C4 fraction produced by the steam cracker is a source of butadiene. Once butadiene is separated the residual, the C4 fraction is mixed with C4 stream from fluid catalytic cracking to separate isobutene for methyl tertiary butyl ether production. The final residual C4 fraction, known as Raffinate II, is the mixture of n- and iso-butanes as well as 1- and 2-butenes. The Raffinate II can be used, for instance, as the feedstock for the metathesis process to produce propylene, or can be used as a pool of petroleum fuels. In case of limited demand for this intermediate, there is another alternative to return it back to the steam-cracking process as the feed [2,3], to obtain products that are more valuable than fuels. 

Nevertheless, the cracking of this feedstock, which may contain tens of percent of unsaturated hydrocarbons, has not been completely studied under industrially relevant conditions yet, although kinetics studies for some representatives of the C4 fraction were published in other context [4,5,6,7,8], which includes the mechanistic studies of 1- and 2-butenes pyrolysis and C4 pyrolysis reactions that are related to their fuel usage and combustion. At the same time, this feedstock deserves special attention, due to its atypical behavior in comparison with conventional saturated streams. The presence of the double bond in a short hydrocarbon chain fundamentally influences not only the reactivity of the feedstock and the composition of pyrolysis products, but also the formation rate of the coke deposits and thus, controls the whole process [9,10]. The whole situation is even more complicated by the fact that C4 hydrocarbon fraction is not commonly processed by steam cracking alone, but in mixtures with heavier alkane materials with the aim to optimize the cracking capacity and to reduce the coke deposition [11,12,13]. Mutual interactions of hydrocarbons that are present in such a mixture are very essential in such situations and they are also generally important in relation to processing feedstocks that are based on renewable materials [14]. 

Special attention of the authors studying co-pyrolysis of C4 hydrocarbons is paid to changes in the conversion of co-pyrolyzed hydrocarbons. The results of these works are summarized in the paper by Shevel’kova et al. [15]. According to this paper, it is possible to discuss the change in conversion in view of the inhibition or acceleration of the decomposition rate of co-pyrolyzed hydrocarbons, where it depends on the concentration and nature of active radicals that are brought into the system by co-pyrolyzed hydrocarbons. The higher the difference among the co-pyrolyzed hydrocarbons in this sense, the higher the monitored effect. If the levels of radicals supplied by the co-pyrolyzed hydrocarbons are similar, then the relevant temperature will be decisive for the nature and size of the mutual effect in the first place, because the temperature provides for the concentration and nature of the active radicals in the mixture. However, concentrations of the individual components in such a mixture play an important role. For instance, if an inhibitor concentration in the feed mixture is low and its influence on the overall level of radicals is negligible, then the inhibition effect will be either weak or non-existent. At the same time, the inhibitor conversion is accelerated under these conditions in the first place. On the other hand, if the inhibitor concentration is high, and the overall level of radicals does not solely differ from the pyrolysis of the inhibitor, then a significant inhibition effect on the co-pyrolyzed hydrocarbon will take place and the opposite acceleration effect will not develop [16]. There is also an alternative processing route involving the catalyzed steam-cracking-process [17], which is referenced here for comparative purposes.

In this study, we proceeded from the above-mentioned general assumptions, but we decided to describe the conversion change not only qualitatively but also quantitatively. Such efforts can be noted also with other authors who are trying to solve the issue by the application of mechanistic or semi-mechanistic models requiring a time-consuming development of the hydrocarbon decomposition reaction network, and even in this case, it is often necessary to work under significantly simplifying assumptions (e.g., [18,19,20,21,22]), where different authors neglect secondary radical reactions for heavier radicals or replace them with a set of substitute molecular reactions. For these reasons, we decided to propose in this study a quantitative evaluation issuing from a regression analysis of experimental data, which allows a generalization of the results from a complete experimental series, while the evaluation is not burdened by excessive a priori assumptions on a concrete reaction network of hydrocarbon decomposition. 

Therefore, this study was aimed at the experimental exploration of different C4 hydrocarbons pyrolysis performances at industrially relevant conditions, including their co-processing with C6+ primary naphtha hydrocarbons in the steam-cracking process, since no such data were published before. The main objective was to propose a suitable regression model capable of describing the interaction of co-pyrolyzed hydrocarbons affecting their conversions and verify the model extrapolation possibilities on independent datasets. 

## 2. Materials and Methods

The experiments were conducted with model mixtures, which always contained a C4 hydrocarbon together with another hydrocarbon with a longer chain (the so-called co-pyrolysis partner). All tested hydrocarbons (summarized in Table 1) were obtained commercially, and their purity exceeded the level of 99% (*w*/*w*). 

### 2.1. Preparation of Model Mixtures

Pre-studies focused on the stability of binary mixtures containing, under normal temperatures, gaseous and liquid hydrocarbons, which showed that the most suitable way of preparing the binary mixtures was in the gaseous phase in a closed vessel. Therefore, the model mixtures were prepared in Tedlar^®^ bags by gasification of C6+ hydrocarbon in the nitrogen inert gas environment enabling its gasification under the laboratory temperature. The resulting mixture was consequently analyzed by gas chromatography (GC), in order to obtain exact concentrations of both components in the mixture and to determine hydrocarbon partial pressures in the mixture. Once the mixture prepared in this way was pyrolyzed, defined quantities of C4 hydrocarbon and nitrogen were added so that a change in the ratio of co-pyrolyzed hydrocarbons was achieved, while the constant partial pressure of the sum of hydrocarbons in the mixture remained preserved. The resulting mixture was then repeatedly analyzed by GC and again pyrolyzed in a pyrolysis gas chromatograph.

### 2.2. Experimental Conditions and Method of Evaluation

The pyrolysis experiments were carried out using the pyrolysis gas chromatograph consisting of a quartz micro-reactor (Shimadzu Pyr-4A, Kyoto, Japan) on-line connected to a series of two gas chromatographs (Shimadzu GC 17-A). This system has been described in detail in our previous studies [23,24]; it provides very reliable results that are suitable for the development of mathematical models of pyrolysis reactions [25,26], which can be consequently used for a very complex evaluation of raw materials for pyrolysis processes [27,28]. 

The experimental conditions were chosen in a way that the achieved conversions would be as close as possible to the values existing in the industry. Therefore, the reactor temperature was set to 820 °C, pressure to 400 kPa, carrier gas flow rate (nitrogen) to 25 Nml/min for co-pyrolysis experiments and 50 NmL/min for pyrolysis of individual hydrocarbons, which corresponded to the sample residence time of about 0.2 s and 0.1 s, respectively. The reactor was equilibrated at specified conditions with the carrier gas flow and then the sample was injected with a pulse of 0.2 mL of gaseous mixture at ambient conditions. The low sample volume and the comparatively high heat capacity of the reactor ensured insignificant temperature change during the injection. For the pyrolysis of individual hydrocarbons, three measurements were carried out and the average value of these three measurements was taken for the final result. Separate measurement values did not show any significant deviations. 

The experimental data were evaluated using the ERA program [29], which is a software application for regression analyses of experimental data. All experiments were evaluated simultaneously and the same parameter values for C6 hydrocarbons were preserved. The hydrocarbon mixtures, where the values of hydrocarbons partial pressure showed deviations higher than 2 kPa from the medium value of all tested mixtures, were eliminated from the evaluation. 

The evaluation in the ERA program consisted in acquisition of parameter optimum values for non-linear regression of experimental data. The ERA program carries out this operation using the method of adaptive random searching with the aim of obtaining the minimum value of the objective function. The objective function was based on weighted least squares method, where the weight coeffcient was automatically set as a reciprocal value of the response variance and response variances were approximated by their residual variances. For the parameter reliability evaluation, confidence limits were calculated at the significance level of 95%. The optimisation procedure and confidence limit calculations are shown in the reference [30]. 

As C4 hydrocarbon may be formed from the C6+ co-pyrolysis partner as an intermediate product, the C4 hydrocarbon conversion was calculated according to the following equation:(1)$$\varsigma_{4}=\frac{w_{4,in}-w_{4,out}-w_{4,k}(1-w_{4,in})}{w_{4,in}}$$
where *ζ*_4_ is the conversion of C4 hydrocarbon in the binary mixture, *w*_4,*in*_ is the mass fraction of C4 hydrocarbon in feedstock, *w*_4,*out*_ is the mass fraction of C4 hydrocarbon in the products, and *w*_4,*k*_ is the mass fraction of C4 hydrocarbon that was formed by the pyrolysis of its co-pyrolysis partner. In this case, *w*_4,*k*_ is the function of the co-pyrolysis partner conversion, and it is obtained by linear regression from the relation of the C4 hydrocarbon mass fraction formed by pyrolysis of pure co-pyrolysis partner on the co-pyrolysis partner conversion. Thus, the equation calculates the net consumption of C4 in the feed that is corrected for the C4 formation from the co-pyrolysis partner. There were no changes in the selectivity of C4 hydrocarbon formation from the co-pyrolysis partner due to the interaction with the C4 hydrocarbon and, in addition, the formation of the co-pyrolysis partner from the C4 hydrocarbon was assumed in the calculation.

## 3. Results

The reaction products of pyrolysis of the model mixtures as well as that of the individual hydrocarbons consisted of hydrogen and a wide range of hydrocarbons. According to the regular industrial practice, the pyrolysis product distribution is shown for hydrogen, C1 to C4 hydrocarbons, and important aromatic hydrocarbons on a component basis, while a group basis was chosen for the other hydrocarbons. The primary data were further processed by the proposed regression model to investigate the co-pyrolysis effects.

### 3.1. Overview of Pyrolysis and Co-Pyrolysis Experimental Results 

As the conversion of the C4 hydrocarbons in the co-pyrolysis process depends on the co-pyrolysis partner conversion (see Equation (1)), the pyrolysis of the pure co-pyrolysis partners was carried out first (see Table 2). 

The differences among the individual hydrocarbons and their yields in pyrolysis products are clearly noticeable. Heptane, as the longest tested hydrocarbon, provides the highest ethylene yields, while 3-methylpentane provides the highest yields of propylene and methane, due to the substitution with methyl that prevents β-splitting with a large yield of ethylene.

In the framework of the co-pyrolysis experiments, the pyrolysis of the mixture of C4 hydrocarbons with n-hexane and cyclohexane was carried out. Table 3 illustrates the results obtained for the 1-butene and n-hexane co-pyrolysis pair. Other pairs were investigated according to a similar design and the primary data are available in the Appendix A.

The experimental data show the pyrolysis product yields are determined by individual hydrocarbon contents in feedstock in the first place; however, the dependence on the hydrocarbon content is not linear, and the mutual influences of the hydrocarbons are thus evident. This is most obvious in the pyrolysis of mixtures containing cyclohexane, where at low contents of isobutane and 1-butene, the ethylene yield is higher than in the pyrolysis of individual cyclohexane, isobutane, and 1-butene.

In addition, the selected unsaturated hydrocarbons were co-pyrolyzed together with n-heptane and 3-methylpentane with the aim to establish the influence of the co-pyrolysis partner structure on the pyrolysis product yields (see Appendix A). The behavior of these hydrocarbons was closer to that of hexane and the monitored influence of co-pyrolysis on yields of the pyrolysis products was thus smaller than with cyclohexane. 

### 3.2. Regression Model Data Processing

The experimental conditions were chosen to achieve relatively high conversions and so that the hydrocarbon conversions would not be significantly influenced by the initiation or termination step. Therefore, the regression model was based on the description of the propagation phase, where the rate-controlling step is a hydrogen abstraction. The active radical and the C-H bond of the hydrocarbon are involved in the hydrogen abstraction. Hence, based on this assumption, the change in the C4 hydrocarbon conversion in time was described by the second order reaction kinetics of the hydrogen abstraction reaction. The equation is proposed as follows:(2)$$\frac{d\varsigma }{d\tau }=k\;a_{H}\;x_{}(1-\varsigma )\;\sum x_{j}a_{j}$$
where *k* is the reaction rate constant, element *a*_H_ expresses the willingness of C4 hydrocarbon for the hydrogen abstraction from various places in the hydrocarbon chain. It is assumed that each C4 hydrocarbon contains a various number of differently active atoms with respect to the C–H bond energy that are parameterized in the mathematical description by a single common parameter *a*_H_, which expresses the sum of contributions from individual hydrogen atoms. This concept was introduced and validated by Karaba [25,26]. *x* is the molar fraction of C4 hydrocarbon, or any hydrocarbon present in the reaction mixture—which is marked by the index *j*. *ζ* is the conversion of C4 hydrocarbon and *a_j_* expresses the activity (quantity) of active radicals in the mixture, regardless of which hydrocarbon in the mixture they originate from. Again, it is assumed here that each hydrocarbon provides various quantities and types of active radicals, the resulting activity of which is expressed by a single parameter for each hydrocarbon. Due to the conversions achieved (see above), pseudo-stationary conditions were assumed, i.e., the concentration of active radicals is not changing with time. 

For the regression analysis in the ERA program, the above-specified equation was formally modified and transferred to the following integral form:(3)$$\varsigma_{}=1-e^{-k\;a_{H}\;t\;\tau \;{\sum x_{j}\;a}_{j}}\;$$
where *t* indicates the residence time and *τ* is the relative time, which assumes the value of zero and one. In order to reduce the number of parameters, the reaction rate constant *k* was evaluated as a part of the parameter characterizing the willingness of the hydrocarbons for the hydrogen abstraction and the activity of radicals formed by the hexane pyrolysis was considered as the unit activity. The data for the regression analysis included the composition parameter *x*_4_ as the independent variable and the conversion as the dependent variable. Such data are available in Table 3 for the 1-butene/n-hexane combination and in supplementary data for other combinations. 

The parameter obtained by the regression analysis was the parameter *a_j_*, which characterized the amount of active radicals provided by the hydrocarbon and the parameter *W*_H_ = *k t a*_H,_ which characterized the willingness of the hydrocarbon for the hydrogen abstraction. Notably, the parameters can be used not only for the comparison of different hydrocarbon behaviors, but they can be used to calculate the reaction rate using Equation (2) and substituting *ka*_H_ = *W*_H_/*t*.

All data from the hexane and cyclohexane co-pyrolysis with all C4 hydrocarbons were evaluated simultaneously. This means that the optimized parameter values are consistent for all tested hydrocarbons. As at least seven experimental points were available for each mixture, more than 100 cases were thus simultaneously evaluated. The regression analysis results are shown in Table 4. 

As a relatively large amount of data were evaluated simultaneously and, at the same time, the number of parameters was reduced to a minimum, we obtained the values of parameters with the confidence intervals at the 95% significance level, which only differ often at a sixth decimal place. Although the presented error value is characteristic only for the relative evaluation of the parameter values *a_j_* and *W*_H_, the relative comparison of parameter values itself, based on what it is possible to compare individual hydrocarbons, was the very purpose of the regression analysis. By achieving such narrow confidence intervals, the reliability of the quantitative comparison was thus proved for tested hydrocarbons. 

The deviation of the calculated values obtained by entering the parameter optimization values to Equation (3) from the experimental data is shown in Figure 1, where the points illustrate the experimentally obtained data, while the lines have been obtained according to Equation (3). In this and the following figures, no error bars for the individual points are reported, as the analytical standard deviation is smaller than the marker size. The error made due to the mixture preparation cannot be calculated from a repeated measurement as the mixture was prepared in approximate proportions and the exact composition was determined for each prepared mixture. The fitting of the experimental data to the model is very good, except for the saturated hydrocarbons, where the molecular reaction effects do not take place, which is indirectly a part of the radical activity parameter. Owing to the fact that the available data for unsaturated hydrocarbons prevail, the program for non-linear regression assumes here a higher conversion due to the molecular reactions. This can be avoided by an evaluation of the saturated hydrocarbons first and separately, and then, during the simultaneous evaluation of all the hydrocarbons, locking the consequently established parameter values, while the remaining values could be additionally optimized. However, this would infringe the validity of the established parameter values as they would not be consistent for all experimentally established values, which consequently, would prevent a mutual comparisons of all hydrocarbons, which was the original purpose of the regression model. At the same time, it could be possible to describe the molecular reactions by adding additional parameters into Equation (3). Nevertheless, this would result in a reduction of the parameter reliability on the one hand, and, in particular, deviating from the objective of comparing individual hydrocarbons on the other hand. 

From the parameter values characterizing the concentration of the active radicals provided by a particular hydrocarbon into the reaction mixture (Table 4, upper section), it is obvious that the most active hydrocarbon—as regards the radical quantity and activity—is hexane. As a result, hexane accelerated the conversion of all the tested hydrocarbons. The degree of such influence can be deducted from the differences in the parameter values of *a_j_* characterizing the radical activities of co-pyrolyzed hydrocarbons. In this way, the greatest difference is between the activities of radicals originating from hexane and isobutene, while a relatively small difference is in the activities of radicals originating from butane and hexane. To this corresponds the change in the C4 hydrocarbon conversion in Figure 1 and Figure 2, as in the case of hexane and isobutene, a large mutual influence of hydrocarbons is evident, while in the second case, the mutual influence is less significant. The hexane presence is relatively significant also for the conversion of 2-butene, even though it can be assumed, based on the conversion achieved in the pure substance pyrolysis, that the conversion of 2-butene would be accelerated by the hexane presence less than that of butane. Especially in this case, it is well-documented that it is not possible to predict with certainty the hydrocarbon behavior in co-pyrolysis only on the basis of the hydrocarbon behavior at its individual pyrolysis. In such cases, the regression analysis represents a tool for a quantitative assessment of the hydrocarbon behavior in the co-pyrolysis process. This is most important in the case of hydrocarbons, where the co-pyrolysis effect is opposite than what could be expected from the results of pure hydrocarbon pyrolysis. For instance, such a case is well-illustrated by the co-pyrolysis of cyclohexane with linear butenes. If pyrolyzed individually, then cyclohexane shows a lower conversion than butenes. Therefore, it could be assumed that cyclohexane will inhibit the conversion of butenes. However, based on the evaluation of the co-pyrolysis results for these hydrocarbons, it has been found that cyclohexane, by contrast, weakly supports the conversion of butenes, and, at the same time, 1-butene and 2-butene support the cyclohexane conversion. This is given by very similar parameter values characterizing the radical activity, which is connected with a similar character and quantities of active radicals supplied by these hydrocarbons into the system. The increase in the conversion of both co-pyrolyzed hydrocarbons over the whole concentration range means that under all ratios of the hydrocarbons, the quantity of the active radicals in the mixture is higher than in other cases, where they are pyrolyzed individually. In the case of cyclohexane, the increase in the conversion of linear butenes is also caused by the additional reactions of the bi-radical that are formed by its initiation. 

For other tested binary mixtures, the situation was clear. As it has been mentioned for the isobutene–hexane mixture, the least active hydrocarbon was isobutene. In a comparison with other hydrocarbons, this hydrocarbon had the lowest values for both parameters. Its conversion thus increases with the growing ratio of hexane and cyclohexane, and this is evident even at low concentrations of C6 hydrocarbon in the mixture. The favorable fact is that the conversion of such a long hydrocarbon, as the active radical donor, is inhibited to a lesser degree than the isobutene conversion supports. 

The situation is also unambiguous in the binary mixtures of saturated C4 hydrocarbons and cyclohexane, where the saturated C4 hydrocarbons accelerated the cyclohexane conversion. This is evident from the diagrams (Figure 2), where the cyclohexane conversion increases with the growing C4 content, as well as from the parameter values of the radical activity it can be concluded that butane and isobutane supplied into the system are more radicals than cyclohexane, since the decomposition of cyclohexane formed a large number of allylic radicals that are more stable than the ethyl and methyl radicals provided by butane, isobutane, or hexane. 

The example of butane and isobutane clearly shows that in spite of large number of active radicals supplied into the system by them, they do not achieve the same conversion levels as linear butenes. In this case, the second characteristic of the tested hydrocarbons takes place: the willingness of the hydrocarbon for the hydrogen abstraction, which is given by the number and character of the C–H bonds in hydrocarbons. In the case of unsaturated hydrocarbons, the effect of a double bond becomes evident, which weakens the C–H bond in the β-position close to the double bond, which results in an increase in their conversion. 

Therefore, it can be said that while the willingness of the hydrocarbon for the hydrogen abstraction furnishes the information about the absolute value of a hydrocarbon conversion, the activity of the formed radicals is informative of a relative change in the conversion, depending on the ratio of individual hydrocarbons.

## 4. Discussion

The proposed model can be used as a prediction tool for predicting pyrolysis yields of C4 hydrocarbons in industrial streams that are processed by steam cracking. In order to verify the extrapolation possibilities, a pyrolysis experiment of C4 hydrocarbons with 3-methylpentane and heptane was conducted so that it would also be possible to study the influence of the structural effects of co-pyrolyzed hydrocarbons to the achieved conversion values, and, by this, also to the parameter values in the proposed model. 

First, a mixture of 3-methylpentane and isobutene was analyzed as, based on the previous results, the largest change in the conversion of C4 hydrocarbon was expected there; therefore, the most accurate optimized parameter as regards the reliability interval width could be obtained. The target value was the *a_j_* parameter value characterizing the activity of radicals supplied to the mixture by 3-methylpentane. 

This value was obtained by entering the *a_j_* and *W_H_* isobutene parameter values (see Table 4) into Equation (3) and consequently conducting the regression analysis of the experimental data for the isobutene–3-methylpentane mixture. In this way, the value of *a*_3*MP*_ = 1.18, within the confidence interval ⟨0.82; 1.81⟩, has been found. In this way, 3-methylpentane supplies the system with a bit more active radicals than hexane and thus accelerates the conversion of C4 hydrocarbons to a greater degree. This is given by the relatively large concentration of very active methyl radicals in the reaction mixture as well as by the ease of the hydrogen abstraction from the tertiary carbon within 3-methylpentane pyrolysis.

The *a_3MP_* parameter value obtained from the isobutene–3-methylpentane mixture was subsequently used for the extrapolation of the behavior of 2-butene in a 3-methylpentane–2-butene mixture, where the experimental data from this mixture were used subsequently, only for an independent verification. The conformity of the experimental data with the data simulated in this way is evident in Figure 3.

The deviation in the conversions obtained by the extrapolation of experimental data was not significantly higher in a comparison with the deviation obtained by the regression analysis of these data. Through this, it has been confirmed that it is possible to use the data obtained in the pyrolysis of the isobutene–3-methylpentane mixture for the evaluation of the influence of 3-methylpentane to the conversion of 2-butene, and the possibility of such evaluations for other C4 hydrocarbons can be assumed as well.

In the second part of our study, a mixture of isobutene and heptane was pyrolyzed and based on the experimental data and the *a_i_* parameter value characterizing the active radical quantity supplied to the mixture by heptane was sought after. It has been found by the regression analysis that the optimum value of the *a_i_* parameter is *a_nC_*_7_ = 1.16, within the confidence interval ⟨0.74; 2.09⟩. This corresponds with the assumption that this longer hydrocarbon supplies a larger abundance of active radicals to the system, which means that heptane will accelerate the conversion of co-pyrolyzed hydrocarbons to a greater degree than hexane or cyclohexane. Even in this case, the obtained parameter value has been verified on an independent dataset, for which a heptane–2-butene mixture was used. The regression analysis results and extrapolation possibilities for the obtained parameter values to another binary mixture are shown in Figure 4. It is evident that in both cases, the conversion trend is well-described. 

## 5. Conclusions

A regression model capable of describing the interaction of co-pyrolyzed hydrocarbons in view of their conversion has been proposed in this study. The proposed model assumes that the tendency of hydrocarbons to abstract hydrogen and the concentration of active radicals in the mixture are responsible for the conversion changes. The latter parameter can fundamentally vary in the co-pyrolysis and therefore, it is not possible to draw a comprehensive conclusion about the behavior of hydrocarbons in co-pyrolysis only based on their behavior in their individual pyrolysis. The proposed model offers a tool for quantitative evaluations of hydrocarbon’s behavior in co-pyrolysis and thus, for carrying out their comparison in view of the achieved conversions. In addition, this model’s advantage also consists of its use as a prediction model, where the cracking simulation of a single mixture provides the possibility to evaluate the conversion of a whole range of mixtures. Based on this fact, the developed model can be regularly used for product yields prediction and planning of the real industry scale co-pyrolysis of C4 fraction and light naphtha.

## Figures and Tables

**Figure 1 materials-16-01418-f001:**
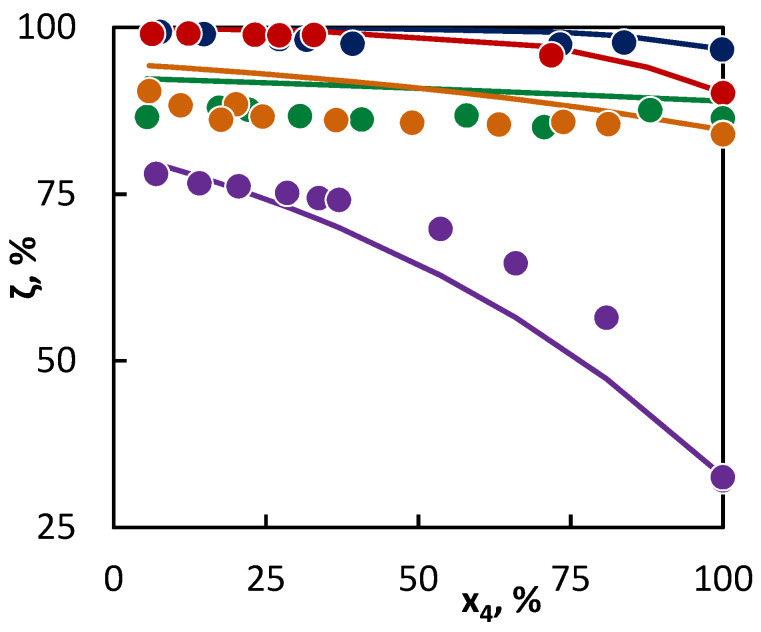
Co-pyrolysis of C4 hydrocarbons and hexane: conversion of C4 hydrocarbons (**●**1-butene; **● **2-butene; **● **isobutane; **●** butane; and **● **isobutene) in dependence on their mole fraction in the binary mixture. Points represent experimental data and lines were obtained from regression analysis.

**Figure 2 materials-16-01418-f002:**
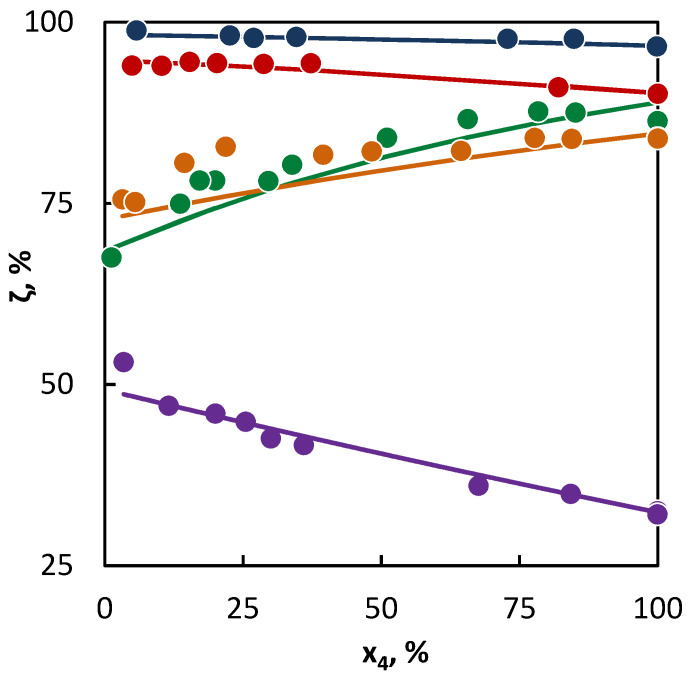
Co-pyrolysis of C4 hydrocarbons and cyclohexane: conversion of C4 hydrocarbons (**●**1-butene; **● **2-butene; **● **isobutane; **●** butane; and **● **isobutene) in dependence on their mole fraction in the binary mixture. Points represent experimental data and lines were obtained from regression analysis.

**Figure 3 materials-16-01418-f003:**
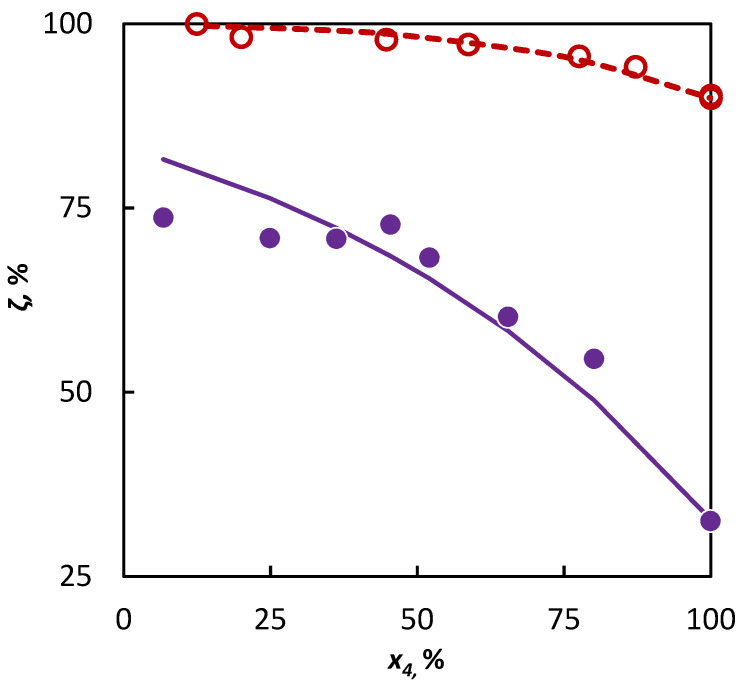
Co-pyrolysis of C4 hydrocarbons and 3-methylpentane: conversion of C4 hydrocarbons (**○ **2-butene and **● **isobutene) in dependence on their mole fraction in the binary mixture. Points represent experimental data; solid line was obtained from regression analysis and dashed line is a simulation.

**Figure 4 materials-16-01418-f004:**
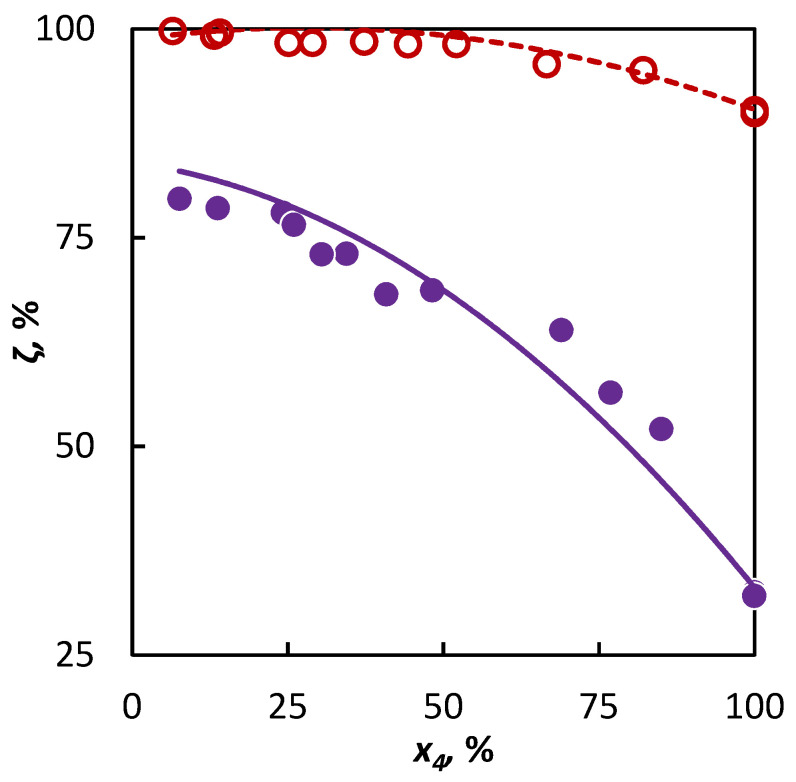
Co-pyrolysis of C4 hydrocarbons and heptane: conversion of C4 hydrocarbons (**○ **2-butene and **● **isobutene) in dependence on their mole fraction in the binary mixture. Points represent experimental data; solid line was obtained from regression analysis and dashed line is a simulation.

**Table 1 materials-16-01418-t001:** List of studied C4 hydrocarbons and C6+ co-pyrolysis partners with their abbreviations.

C4 Hydrocarbon		Co-Pyrolysis Partner	
Name	ID	Name	ID
butane	nC4	hexane	nC6
2-methylpropane	iC4	cyclohexane	cC6
but-1-ene	1 = C4	3-methylpentane	3MP
(Z)-but-2-ene	2 = C4	heptane	nC7
2-methylpropene	i = C4		

**Table 2 materials-16-01418-t002:** Pyrolysis of pure co-pyrolysis partners at 820 °C and carrier gas flow rate of 25 mL/min and 50 mL/min. Composition (*w*/*w*) of reaction mixture after pyrolysis (in columns: hexane; cyclohexane; 3-methylpentane; and heptane).

Feedstock	nC6	nC6	cC6	cC6	3-MP	3-MP	nC7	nC7
Flow Rate, mL/min	25	50	25	50	25	50	25	50
Hydrogen	1.1	1.0	1.1	0.8	0.5	0.7	0.9	1.2
Methane	12.4	10.5	2.5	1.7	17.4	14.1	12.0	9.1
Ethane	2.4	2.3	0.5	0.4	4.1	4.8	2.3	2.3
Ethylene	57.4	50.1	29.3	23.2	39.0	35.4	61.3	56.3
Propane	0.4	0.5	tr. ^1^	tr.	0.4	0.5	0.4	0.4
Propylene	13.8	15.8	4.6	4.0	17.5	18.5	12.7	14.1
Acethylene	0.8	0.6	1.1	0.7	1.0	1.1	0.7	0.7
Isobutane	tr.	0.2	0.1	0.2	tr.	tr.	0.2	0.3
Propadiene	0.3	tr.	0.1	0.1	0.7	0.8	tr.	tr.
n-Butane	tr.	tr.	tr.	tr.	tr.	tr.	tr.	tr.
(E)-2-Butene	0.2	0.2	0.1	0.1	0.8	1.3	0.2	0.2
1-Butene	1.5	3.5	0.3	0.5	1.9	3.2	1.1	3.0
Isobutene	0.1	0.1	tr.	tr.	1.9	2.1	0.1	0.1
(Z)-2-Butene	0.2	0.2	0.1	0.1	0.7	1.1	0.1	0.2
Propyne	0.4	0.2	0.3	0.2	0.8	0.9	0.3	0.3
Butadiene	4.2	4.4	22.9	20.5	7.1	7.0	3.9	4.2
Feedstock	2.7	8.5	27.7	40.7	1.0	3.9	1.5	5.6
Cyklopentadiene	0.4	0.3	1.5	tr.	1.2	1.3	0.3	0.3
Other NA ^2^ C5–C6	0.6	0.9	2.1	3.6	1.1	1.7	0.4	0.7
Benzene	1.1	0.5	4.1	2.4	1.9	0.8	1.1	0.7
Toluene	0.1	0.1	0.5	0.3	0.4	0.3	0.1	0.1
Ethylbenzene	tr.	tr.	tr.	tr.	tr.	tr.	tr.	tr.
m- +p-Xylene	tr.	tr.	tr.	tr.	tr.	tr.	tr.	tr.
Styrene+o-Xylene	tr.	tr.	0.1	tr.	tr.	tr.	tr.	tr.
Naftalene	tr.	tr.	tr.	tr.	tr.	tr.	tr.	tr.
Other C7–C12	tr.	tr.	0.7	0.5	0.3	0.2	0.3	0.3
C12+	0.1	0.1	0.2	0.2	0.1	0.1	0.1	0.1

^1^ tr. means no product or only traces of that product were detected below the quantification limit. ^2^ Non-aromatics.

**Table 3 materials-16-01418-t003:** Co-pyrolysis of 1-butene and n-hexane at 820 °C. Composition of reaction mixture after co-pyrolysis (in columns: content of 1-butene in the feedstock).

1-Butene, %	5.0	10.0	19.2	22.7	29.3	74.4	85.3	100
Hydrogen	0.7	0.7	0.6	0.5	0.5	tr.	Tr.	Tr.
Methane	13.1	12.9	13.2	13.1	12.5	13.8	13.6	13.2
Ethane	2.6	2.6	2.3	2.3	2.2	2.3	2.1	2.3
Ethylene	57.1	56.4	51.3	51.0	48.9	34.8	31.5	26.5
Propane	0.3	0.3	0.3	0.3	0.3	0.2	0.1	0.1
Propylene	13.8	14.0	15.3	15.4	16.0	17.9	18.4	20.3
Acethylene	1.1	1.1	0.8	0.9	1.0	1.3	1.5	1.3
Isobutane	tr.	Tr.	Tr.	Tr.	Tr.	Tr.	Tr.	Tr.
Propadiene	0.3	0.3	0.3	0.3	0.4	0.6	0.6	0.7
n-Butane	tr.	Tr.	Tr.	Tr.	Tr.	0.1	0.1	0.1
*€*-2-Butene	0.2	0.2	0.2	0.3	0.3	0.5	0.5	0.7
1-Butene	1.5	1.5	1.7	1.8	2.1	2.1	2.1	3.4
Isobutene	0.1	0.1	0.2	0.2	0.1	0.5	0.4	0.3
(*Z*)-2-Butene	0.2	0.2	0.2	0.2	0.3	0.4	0.5	0.6
Propyne	0.4	0.5	0.4	0.4	0.5	0.9	1.0	1.1
Butadiene	4.5	4.8	6.2	6.5	7.2	12.4	13.6	15.3
Hexane	2.0	2.1	2.3	2.2	2.6	0.8	0.5	tr.
Cyclopentadiene	0.3	0.3	0.6	0.6	0.8	1.3	1.6	1.5
Other NA C5–C6	0.6	0.6	0.8	0.9	1.1	1.7	1.9	3.1
Benzene	1.1	1.2	2.1	2.0	2.3	5.2	6.5	6.8
Toluene	0.1	0.1	0.4	0.4	0.4	1.0	1.2	1.2
Ethylbenzene	tr.	tr.	tr.	tr.	tr.	0.1	0.1	tr.
m- +p-Xylene	tr.	tr.	tr.	tr.	tr.	0.1	0.1	tr.
Styrene+o-Xylene	tr.	tr.	tr.	tr.	tr.	0.5	0.5	tr.
Naphthalene	tr.	tr.	tr.	tr.	tr.	0.1	0.1	tr.
Other C7–C12	tr.	tr.	0.6	0.5	0.4	1.1	0.9	1.5
C12+	tr.	tr.	0.1	0.1	0.2	0.4	0.4	0.3

**Table 4 materials-16-01418-t004:** Optimized parameter values obtained from regression analysis of experimental data according to Equation (3).

Parameter	Hydrocarbon	Value	Confidence Interval 95%
*a_j_*	nC6	1	-
	cC6	0,468,368	⟨0,468,367; 0,468,369⟩
	i = C4	0,237,720	⟨0,237,719; 0,237,720⟩
	1 = C4	0,403,451	⟨0,403,446; 0,403,452⟩
	2 = C4	0,365,343	⟨0,365,341; 0,365,344⟩
	nC4	0,836,189	⟨0,836,187; 0,836,196⟩
	iC4	0,679,227	⟨0,679,223; 0,679,228⟩
*W* _H_	i = C4	1,660,758	⟨1,660,755; 1,660,758⟩
	1 = C4	8,423,175	⟨8,423,154; 8,423,189⟩
	2 = C4	6,278,456	⟨6,278,435; 6,278,507⟩
	nC4	2,577,345	⟨2,577,331; 2,577,352⟩
	iC4	2,786,564	⟨2,786,560; 2,786,572⟩

## Data Availability

Data are contained within the article and Appendix A.

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
