# Peer review of "Co-Pyrolysis of Unsaturated C4 and Saturated C6+ Hydrocarbons—An Experimental Study to Evaluate Steam-Cracking Performance"

_materials, 2023, doi:10.3390/ma16041418_

Round 1

Reviewer 1 Report

This paper had been revised and I though it could be accepted in the present form. 

Author Response

Review Comment 1: abstract should be promoted according to the objective.

We thank the reviewer for pointing out at this problem. The abstract used unclear term of „hydrocarbons with longer chain“. We explained that this means C6+ hydrocarbons and also why they were selected.

Review Comment 2: In the introduction parts, the author should summarize the results of previous researches and clarify the novelty of this paper.

Based on this comment, we specifically added the description of previous studies 4-8. We also expanded last paragraph of the introduction section to clarify the novelty and further specify the objectives.

Review Comment 3: The basis of change in the C4 hydrocarbon conversion (Eq. 2) should be clarified further

In addition to detailed description of symbols in this equation, we also added the general rationale for this equation which is based on the second order kinetics of the hydrogen abstraction. We also added references for our previous papers using similar concept. We also added another explanation on the request of another reviewer.

Review Comment 4: In the results parts, in Fig 1 and Fig. 2, the errors of experimental data and data from regression analysis should by illuminated further.

We explained that “In this and the following figures, no error bars for the individual points are reported, as the analytical standard deviation is smaller than the marker size. The error made due to the mixture preparation cannot be calculated from repeated measurement as the mixture was prepared in approximate proportions and the exact composition were determined for each prepared mixture.“ The residual variance of the description by the model which is apparent from the figures is attributed to the regression model simplification and it is reflected by the confidence intervals.

Review Comment 5: There were some study investigated the characterizations of co-pyrolysis and obtained interesting results. In the discussion parts, the results of similar researches should be compared to promote the quality of this paper.

Unfortunately, to our best knowledge, there are no published results on the c4/c6 copyrolysis at even roughly similar conditions. Therefore, such comparison is not possible.

Reviewer 2 Report

In this paper, the steam co-pyrolysis process of C4 and above hydrocarbons was studied, and the corresponding pyrolysis model was obtained. The research purpose of this paper is clear and innovative, but there are also the following problems:

(1) In the introduction, the author should clearly point out the innovation of this paper and the work on the novelty of this field or what engineering problems or academic innovations have been solved. 

(2) The calculation of conversion rate in formula 1 is not rigorous enough. Although the author has explained the equation, is there any basis for this treatment. Cracking is a very complex equilibrium reaction process. Whether the author's current treatment method is reasonable or not needs further supplement.

(3) In section 2.2, although the author explained the experimental conditions, but did not explain the temperature of raw materials, the temperature of carrier gas and the velocity of injection. The reaction time involved in this paper is very short, so the above data has a great impact on the reaction and needs to be supplemented.

4The proposed model can be used as a prediction tool for predicting pyrolysis yields of C4 hydrocarbons in industrial streams processed by steam cracking. Only one reaction time was done in the experiment. The residence time has a great influence on the yield of pyrolysis products

(5) For reactor design, more attention is paid to kinetics, but no corresponding kinetic data of co-pyrolysis is obtained in this paper. Although it is difficult, please try to give an empirical kinetic parameter according to the lumped kinetic method and reflect the role of co-pyrolysis

(6)The content of the manuscript is a little small, only four Fig.

Author Response

(1) In the introduction, the author should clearly point out the innovation of this paper and the work on the novelty of this field or what engineering problems or academic innovations have been solved. 

Based on this comment and the comments of other reviewers, we expanded last paragraph of the introduction section to clarify the novelty and further specify the objectives.

(2) The calculation of conversion rate in formula 1 is not rigorous enough. Although the author has explained the equation, is there any basis for this treatment. Cracking is a very complex equilibrium reaction process. Whether the author's current treatment method is reasonable or not needs further supplement.

The cracking reaction at such conditions is governed mainly by kinetics, as the equilibrium concentration of the starting material is low. The equilibrium effect was therefore neglected but the effect of consecutive reactions producing additional C4 from the C6 partner had to be accounted for. Therefore, the conversion equation 1 was  corrected for this formation. The additional description was added to the article. We also added the general rationale for equation 2 which is based on the second order kinetics of the hydrogen abstraction. We also added references for our previous papers using similar concept.

(3) In section 2.2, although the author explained the experimental conditions, but did not explain the temperature of raw materials, the temperature of carrier gas and the velocity of injection. The reaction time involved in this paper is very short, so the above data has a great impact on the reaction and needs to be supplemented.

We thank to the reviewer for noticing that the information is not present. We referred for the experimental details to our previous work and simplified the description, so that the information about pulse injection of the sample was completely omitted. Therefore, we added necessary details to the experimental section.

(4) The proposed model can be used as a prediction tool for predicting pyrolysis yields of C4 hydrocarbons in industrial streams processed by steam cracking. Only one reaction time was done in the experiment. The residence time has a great influence on the yield of pyrolysis products

We can only agree with this remark. The aim of this study was to compare the different C4s and to quantify the co-pyrolysis effects, as they are difficult to model accurately due to the substantial impact of secondary reactions. The effect of time, temperature and dilution can be well described by the mathematical modelling as we demonstrated in ref. 18 or 25 as well as did other authors. The study, as it is completed include about one year of measurements and cannot be expanded further at this time.

(5) For reactor design, more attention is paid to kinetics, but no corresponding kinetic data of co-pyrolysis is obtained in this paper. Although it is difficult, please try to give an empirical kinetic parameter according to the lumped kinetic method and reflect the role of co-pyrolysis

In the light of this comment, we described the interpretation of calculated parameters to provide total reaction rate of the c4 hydrocarbon.

(6)The content of the manuscript is a little small, only four Fig.

We commented in (4) that this study was quite demanding experimentally. Also, there are 13 additional tables of data in supplementary material, which we did not included in the paper directly for clarity. Thus, we cannot agree with the statement that this paper content is small.

Reviewer 3 Report

Jiří Petrů investigated the Co-pyrolysis experiments aiming at the comparison of C4 hydrocarbons were performed 12 in a laboratory micro-pyrolysis reactor under standardized conditions: 820 °C, 400 kPa and 0.2 s 13 residence time in the reaction zone. C4 hydrocarbons were co-pyrolyzed with different co-pyrolysis 14 partners containing longer hydrocarbon chain to study the influence of the co-pyrolysis partner 15 structure on the behavior of C4 hydrocarbons, especially regarding yields of the pyrolysis products 16 and the conversion of C4 hydrocarbons.

I recommend the manuscript for publication after the following corrections

1. Enrich your abstract with statistical conclusive data. 

2. At page 2, Line 82 bulk of references has been used, would be better to explain these references separately. 

3. The pyrolysis temperature is 820 C , which is too high. have you optimized the temperature. if yes then please add detail of optimized data.

4.  The pyrolysis temperature is 820 C , which is too high, can temperature can be reduced by changing the conditions or by applying a suitable catalyst.?

Author Response

  1. Enrich your abstract with statistical conclusive data. 

Based on this comment, we extensively reworked the abstract.

  1. At page 2, Line 82 bulk of references has been used, would be better to explain these references separately. 

We explained the general simplifications that the cited authors made.

  1. The pyrolysis temperature is 820 C , which is too high. have you optimized the temperature. if yes then please add detail of optimized data.

The steam cracking of C3-C8 hydrocarbons is carried out at temperatures around 840 °C in modern reactors. We used 820 °C as the maximum temperature allowed by our experimental device. It should be noted that this is the peak temperature of the temperature profile. The detailed data on the profile and the apparatus are available e.g. in ref. 24.

  1. The pyrolysis temperature is 820 C , which is too high, can temperature can be reduced by changing the conditions or by applying a suitable catalyst.?

The temperature can be lowered by using suitable catalyst, but that would relate the research to a catalytic pyrolysis process. Our objective was relate the research to purely thermal pyrolysis = steam cracking, which is industrially preferred process due to equilibrium, capacity and technical reasons.

Reviewer 4 Report

The work aims at quantitative description of the hydrocarbons behavior in the co-pyrolysis process by the regression model developed on the base of experimental data obtained at the laboratory scale conditions. The model describes the process with high accuracy. The manuscript is well structured, figures are well-defined, the methods of hydrocarbon mixtures characterization are relevant. The results of the work are valuable from the practical point of view for prediction the yields of the products in the steam cracking process.

Some comments concerning the work include:

1. The authors do not explain why the particular co-pyrolysis partners (hexane, cyclohexane, 3-methylpentane, heptane) were chosen (Table 1).

2. It’s not clear how the willingness of hydrocarbons for the hydrogen abstraction and the activity of radicals were estimated (lines 221-224).

I therefore recommend accepting the manuscript after minor revision.

Author Response

  1. The authors do not explain why the particular co-pyrolysis partners (hexane, cyclohexane, 3-methylpentane, heptane) were chosen (Table 1).

We included the information that C6+ hydrocarbons were selected as the typical representatives of primary naphtha in the abstract and the introduction. The primary naphtha is the suggested industrial co-pyrolysis partner.

  1. It’s not clear how the willingness of hydrocarbons for the hydrogen abstraction and the activity of radicals were estimated (lines 221-224).

Unfortunately, we described only the regression equation in the paper and the regression data description was missing. Based on this comment, we specified the data used for the regression in the text.